# A Review on Microfluidics: An Aid to Assisted Reproductive Technology

**DOI:** 10.3390/molecules26144354

**Published:** 2021-07-19

**Authors:** Anand Baby Alias, Hong-Yuan Huang, Da-Jeng Yao

**Affiliations:** 1Institute of NanoEngineering and MicroSystems, National Tsing Hua University, Hsinchu 30013, Taiwan; alias.anand@gmail.com; 2Department of Obstetrics and Gynecology, Chang Gung Memorial Hospital, Taoyuan 33305, Taiwan; 3Department of Obstetrics and Gynecology, Chang Gung University and College of Medicine, Taoyuan 33305, Taiwan; 4Department of Power Mechanical Engineering, National Tsing Hua University, Hsinchu 30013, Taiwan

**Keywords:** droplet microfluidics, assisted reproductive technology, fertilization in vitro, sperm and oocyte, embryo culture

## Abstract

Infertility is a state of the male or female reproductive system that is defined as the failure to achieve pregnancy even after 12 or more months of regular unprotected sexual intercourse. Assisted reproductive technology (ART) plays a crucial role in addressing infertility. Various ART are now available for infertile couples. Fertilization in vitro (IVF), intracytoplasmic sperm injection (ICSI) and intrauterine insemination (IUI) are the most common techniques in this regard. Various microfluidic technologies can incorporate various ART procedures such as embryo and gamete (sperm and oocyte) analysis, sorting, manipulation, culture and monitoring. Hence, this review intends to summarize the current knowledge about the application of this approach towards cell biology to enhance ART.

## 1. Introduction

About 7% of couples worldwide suffer from infertility conditions [1]. Assisted reproductive technology (ART) includes all treatments and medical procedures that include the handling of eggs and embryos to address infertility. Through ART, part of infertility in women can be resolved by manual processing. In recent reproductive medicine technology, various methods are available for infertile couples. ART can increase the genetic quality of livestock, produce transgenic animals, assisting in cloning and artificial insemination, reduce disease transmission, preserve endangered germplasm, produce chimeric animals for disease research and treat infertility. It can improve fertility through artificial treatments, including fertilization in vitro (IVF) [2], gamete intra-fallopian transfer (GIFT), pronuclear-stage tubal transfer (PROST), tubal embryo transfer (TET), zygote intra-fallopian transfer (ZIFT), intrauterine insemination (IUI) and intracytoplasmic sperm injection (ICSI) [3]. ART makes a continuing endeavor to improve outcomes and to optimize efficiency. The most typical ART methods are fertilization in vitro (IVF) and intracytoplasmic injection of sperm (ICSI) [4].

In the case of the male reproductive system, infertility is most commonly caused by problems of semen ejection, absence or low levels of sperm, or abnormal morphology or motility of sperm. In the female reproductive system, infertility might be caused by abnormalities of the ovaries, uterus, fallopian tubes and endocrine system. ICSI is a technique in which a single sperm is injected directly into an oocyte for fertilization, which might cause unexpected damage due to manipulation of the sperm and oocytes [5].

Fertilization in vitro (IVF) became an exciting scientific achievement of the twentieth century that made a significant impact on human lives [6]. IVF involves procedures, in a complicated series, that are used to treat infertility and genetic problems that assist in childbirth. It is an effective form of ART. The rates of success of fertilization in vitro and embryo transfer in humans have risen dramatically, since Louise Brown was born in 1978, through improvements in ovulation induction medications and regimens and progress in culture technology, including culture media [7]. In the face of widening ART applications, there are still compelling issues associated with it, such as small rates of implantation and large rates of multiple pregnancy [8].

IVF treatment is an emotional and physical burden for women and their partners. Research results show that couples that set foot in an IVF-treatment program are, in general, psychologically well-adjusted [9]. Chiware et al. [10] investigated the current availability of IVF in low- or middle-income countries (LMIC).

### 1.1. Conventional IVF & ICSI 

During IVF, mature oocytes are retrieved from a patient’s ovaries and fertilized with a sperm in the laboratory. The fertilized oocyte is subsequently implanted into the patient’s uterus. Current IVF procedures include selecting motile sperm, retrieval and processing of oocytes, insemination, embryo culture in vitro and cryopreservation. Two main approaches of insemination in a laboratory are conventional IVF and intracytoplasmic sperm injection (ICSI). During conventional IVF, every oocyte is placed in an oil-covered micro drop in a culture dish with an optimal concentration of sperm, and fertilization occurs when one sperm penetrates the oocyte in a natural way. In contrast, in ICSI, a single sperm is injected directly into the oocyte’s cytoplasm through a thin needle. ICSI is the presiding treatment for almost all forms of male infertility [11]. There is no need to undertake ICSI in most cases unless there is a male factor or history of previous failure in fertilization, but the truth is that many groups use ICSI in women who have poor prognosis, women who are undergoing PGS, or for fertilization of vitrified oocytes. The results show that ICSI is used regularly in a laboratory for >70% of cases. Two essential steps in the early stages of IVF could be standardized using robotics: removing cumulus cells surrounding the oocyte at retrieval and sperm injection into the cytoplasm [12].

An early insightful opinion on miniaturization and mini-machines for biological applications was presented by Professor Richard P. Feynman (Nobel Laureate in Physics, 1965) [13].

### 1.2. Droplet Microfluidics

Droplet-based microfluidics manipulates and controls individual sections of fluid, rather than the bulk fluid flow. In continuous microfluidic flow, the fluid mixing is dominated by molecular diffusion, and hence cross contamination may occur. The main attractions of droplet based microfluidics include a reduction in sample/reagents, faster reaction times, and increased control over dispersion, mixing, and separation [14]. 

Droplet-based microfluidic systems are compatible with many chemical and biological reagents and are capable of performing a variety of digital fluidic techniques as programmable and reconfigurable operations. This technology has the potential to provide novel solutions to today’s biomedical engineering challenges for advanced diagnostics. Droplet-based microfluidic systems are sometimes referred to as digital microfluidics. This term aims to emphasize the utilization of discrete and distinct volumes of fluids and to contrast these with the continuous nature of other systems. Digital or droplet-based microfluidics involves the generation and manipulation of discrete droplets inside micro devices. This method produces highly monodisperse droplets in the nanometer to micrometer diameter range, at rates of up to twenty thousand per second, used as micro reactors in the nano-liter to femto-liter range [15]. Based on the fabrication and handling technique, the field itself can be further divided into two groups: (i) emulsion- based droplet microfluidics and (ii) digital microfluidics.

#### 1.2.1. Emulsion-Based Droplet Microfluidics

In emulsion-based droplet microfluidics, micro droplets are formed from the interaction of two immiscible fluids (such as oil and water-in continuous and dispersed phases, respectively). For ART application, this branch of microfluidics is very similar to what embryologists perform in clinical IVF, i.e., encapsulating the gamete in microdroplets covered with an oil overlay. Moreover, the technology can overcome many handling difficulties involving microdroplets in such clinical settings [14]. 

#### 1.2.2. Digital Microfluidics

Digital microfluidics (DMF) is a popular technique for diverse applications. Digital microfluidics is an emerging and very promising field in droplet-based microfluidics. Here, individual droplets (either in open or confined channels) can be precisely manipulated by non-contact forces such as electrical, magnetic or thermal ones. The non-contact forces are of great benefit, especially for ART and IVF applications. DMF has a range of discrete fluidic operations for the manipulation of nano-litre to microliter droplets containing samples and reagents. Here, electric potential has been applied in series to array of patterned electrodes which are coated with a hydrophobic insulator. DMF can address each reagent individually with no need for complicated networks of tubing or micro-valves; it has the ability to control liquids relative to solids with no risk of clogging, and it is compatible with a large range of volumes, making it useful for preparative-scale sample handling [16]. The DMF device has been widely applied for biomedical applications. The advantages of DMF include minimized consumption of reagents, rapid reactions and modest cost (e.g., DNA-based applications, protein-based applications, cell-based applications and clinical applications [17,18,19,20,21,22,23,24]). A review of analogue and digital microfluidics is shown in Figure 1 [25]. Electro-wetting-on-dielectric (EWOD) is an example of a DMF platform.

##### Electro-Wetting-On-Dielectric

EWOD is a method to manipulate discrete droplets on a solid (usually hydrophobic) surface, in a programmable manner, using electrodes made of a conductor such as indium tin oxide (ITO). At least one of the surfaces that is in contact with the liquid is insulated with a layer of dielectric, e.g., Parylene or SU8, and a layer of a hydrophobic material, e.g., Teflon or Cytop. The electrode pattern defines the desired droplet path. Therefore, by arranging the electrodes in the desired way, a variety of parallel or sequential analyses can be accomplished [14].

### 1.3. Microfluidics towards ART

Microfluidic techniques have become a powerful tool that can closely recreate the physiological conditions of organ systems in vivo [12]. Microfluidics can be considered as both a science and a technology. Scientifically, it is the study of fluid behavior at a sub-microliter level and the investigation of its application to the fields of cell biology, chemistry, genetics, molecular biology and medicine [26]. Technologically, it relates to applications within analytics and diagnostics, cell biology and single-cell genomics [27]. Microfluidics broadly represents a multi- and trans-disciplinary field of study comprising engineering, physics, chemistry, biology and biotechnology. Systems with small fluid volumes used to multiplex, to automate, to integrate and to facilitate cell manipulation and high-throughput analysis or screening can be designed for practical applications [28]. There are at least two microfluidics—of mechanical and biochemical characteristics—that can influence mammalian gamete and preimplantation embryo biology. Microfluidic mechanical and biochemical characteristics might have practical and technical applications towards ART [28]. Microfluidic devices for biological assays perform analysis with efficient throughput; their advantages include the use of samples or reagents in small amounts and the analysis of single cells [29,30,31].

Fluids at the microscale are likely to suffer from forces that are typically insignificant at scales present in our everyday lives. Fluid at the scale of our typical environment is turbulent (undergoing irregular fluctuations), which leads to a particle within a stream of fluid moving in an unpredictable pattern. Turbulent flow depends on specific fluid characteristics (viscosity, density and velocity) and the geometry of the channel, leading to the calculation of a value known as Reynold’s number. As the scale of the channel reaches micro-meter levels, the Reynold’s number decreases. A decrease in Reynold’s number below a threshold value brings the fluid flow in a laminar fashion so that the flow within a channel becomes streamlined and predictable [32]. Some of the advantages of utilizing a microfluidics platform include scalability, increased automation, and reduced turnaround times for male infertility diagnosis and treatment [33].

#### Basic Functions of a Microfluidic Bio Chip

The fabrication process of a PDMS-based microfluidic device is explained in [34]. Microfluidic systems serve to handle gametes, mature oocytes, culture embryos, and to perform other basic procedures in a microenvironment that closely imitates conditions in vivo [35]. To perform the above stages, a microfluidic biomedical chip must integrate four essential functions—generation of a droplet of an emulsion, sorting, expansion and restoration. A microfluidic biomedical chip, with a PDMS substrate that has all these essential functions, employed in a mouse embryo system to assess reproductive medicine, is explained in [36]. Here, a droplet carries an embryo, proceeds towards an enlargement area, and another droplet without an embryo is abandoned and directed towards the waste area outlet using a manual sorting method.

A PDMS flow-focusing microfluidic device which was used to generate microliter medium-in-oil droplets is explained in [37]. The device contains two inlets for the continuous injection of oil and medium with oocytes. The volume of micro-droplets was controlled by adjusting the flow rates of both oil and medium inlets. A single mouse oocyte was encapsulated in each dispersed micro-droplet for further in vitro fertilization (IVF) research.

### 1.4. Microfluidic IVF

The current research to regulate various stages of assisted reproductive programs with microfluidic techniques, with their benefits and shortcomings, is explained in [11]. IVF is a technique in which sperm and oocytes are conventionally inseminated and cultured in specialized laboratory conditions. In a traditional IVF operation, unnecessary damage to oocytes occurs on account of the di-electrophoretic force manipulation of the sperm and oocytes [38]. Microfluidic chips might have multiple layers of varied materials for a specific purpose [39,40], i.e., an elastomer bound to glass for a product with increased rigidity. Polydimethylsiloxane (PDMS) is a popular elastomer material of one kind for the fabrication of microfluidic chips. As PDMS is cheap and possesses gas permeability and biocompatibility, most microfluidic devices related to biological research are made of PDMS [41]. Some studies, concerned with microfluidic application in the IVF field, were performed to evaluate the real biocompatibility of the materials required to build a device [42,43,44,45]; these initial studies were performed using mouse embryos [45] and pig oocytes [46]. The results showed that many materials such as silicon nitrate, silicon oxide, borosilicate glass, chromium, gold, titanium and polydimethylsiloxane (PDMS) have no negative impact on embryo development [42,45,46,47].

## 2. Microfluidics in Sperm Sorting

Male factors are responsible for about 50% of infertilities among couples [48], which means that male factors contribute to infertility at approximately the same rate as female factors. The rate of success of IVF might improve with the use of a microfluidic sperm-sorting chip for sperm selection in male infertility [49]. However, accumulated data analyzing a relation between conventional sperm analysis (sperm concentration, motility and morphology) and clinical results show that these male factors are not always predictive of the performance of ART [50]. The purpose of ART is to fertilize the best oocyte with the best sperm [51], which makes the selection of sperm to assist reproduction essential. The efficiency of sorting live sperm can be enhanced with a biomedical microfluidic chip to avoid sperm damage [52]. As the presence of non-motile or damaged sperm, sperm fragments and other debris adversely affects the quality of semen [35], the sorting of sperm is essential. It has been confirmed that microfluidics can help to isolate human sperm efficiently [2]. The sperm penetration assay (SPA) is an evaluation of sperm fertility potential [53]. A large amount of highly active sperm is critical for successful ART operations. In human IVF, one objective is the isolation of sperm of sufficient quality and quantity to facilitate fertilization with either conventional insemination (IVF/CI) or ICSI (IVF/ICSI) [26]. Attempts were made to develop sperm isolation devices without centrifugation [54,55,56]. In the early to mid-1990s [57,58], silicon/glass microfluidic devices were made and tested as diagnostic tools to evaluate sperm motility.

Several microfluidic devices have been explored for sperm diagnosis [57,59,60]. A polydimethylsiloxane (PDMS) microfluidic biochip for sperm sorting is shown in Figure 2a [61]. This method collects highly active sperm from semen with the characteristic of laminar flow. A syringe pump with a small flow rate is combined with a microfluidic chip. Laminar flow is generated on a microscale and applied to sperm samples of varied concentration and mobility. A ‘flow cytometry analysis’of a PDMS microfluidic biochip was performed to verify the viability of sorted sperm from semen with the characteristic of laminar flow [3]. The result is more significant by at least 10% for both regular and oligozoospermia samples. Huang et al. [61] explains a microfluidic system with three inlets and outlets for sorting motile sperm. The laminar flow characteristics have three parallel flows within a microfluidic chip. Only motile sperm can break through the boundary layer from their original laminar flow through the main channel, because of their free-swimming property. This mechanism can, thus, separate motile sperm from non-motile sperm and other particles. The velocity of the sperm is typically 70 μm/s. The motile sperm from the upper and lower inlets in the explained device can break through the stream boundary to the collection reservoir. Syringe pumps controlling the flow rate within 50–100 μm/s led to a smaller velocity and exhibited long-term stability. The method enhanced efficiency by over 90%. The fluorescent images from CFD simulation to analyze the 3D flow-field phenomenon for sperm sorting over a microfluidic chip shows that the laminar streams are the same as for the CFD simulation. The microfluidic system was designed to separate motile sperm according to a design, whereas non-motile spermatozoa and debris flow along their initial streamlines and exit through outlet. In contrast, motile spermatozoa have an opportunity to swim into a parallel stream and exits through a separate outlet.

A microfluidic sperm-sorting device fabricated with soft photolithography, with the PDMS flow channel substrate prepared by mixing PDMS with a curing agent on the mold at a ratio of 10:1 [62], appears in Figure 2b. A comparative study between a COMSOL multi-physics simulation and modeling software application, and the design of an experimental (DOE) method, was performed. The comparison between the simulation and experimental results gave an accuracy 97.28%. An average accuracy 95.33% was achieved by the microfluidic chip in the sorting of live and dead sperm.

Wu et al [63] explains, in the context of a microfluidic system, a ‘flowing upstream sperm sorter’ (FUSS) device that is designed to replicate the selection mechanism found in the female body when sperm swim into the uterus. This system can rapidly (in 15 min) and gently sort sperm according to their motilities by a flow-upstream sorting method that involves the retarding of the flow field. Figure 2c [63] shows their procedure for the separation of motile sperm. 

A hydrophilic PDMS surface can also be suggested for microfluidic channels for creating a stable and hydrophilic surface using PEGMA bonded with PDMS [64]. This hydrophilic surface of PDMS coated with PEG-MA has a contact angle of 35.97° and maintains its stability for at least three days. This modified surface allows the development of a portable and reliable microfluidic system to improve sperm sorting.

The movement of sperm due to a gradient of the chemoattractant concentration is one of the most important factors facilitating the navigation of sperm toward oocyte in a female reproductive tract. This process is termed as sperm chemotaxis [14].

The conventional preparation of sperm for assisted reproduction is criticized for its centrifugation steps. The centrifugation either recovers motile sperm in variable proportions or increases the probability of damage to sperm DNA [64]. Sperm DNA damage is associated with a significantly increased risk of pregnancy loss after IVF and ICSI [65].

Taxis is a behavioral response of a cell or an organism to external stimulus. ‘Rheotaxis’ is the ability of a sperm cell to orient in the direction of the flow and swim against it. A sperm-sorting microfluidic device that exploits the rheotaxis of sperm is explained in [66]. A spiral microfluidic channel for the separation of sperm from blood cells is explained in [67]. The recent developments in microfluidics in the field of sperm sorting are explained in [68]. 

## 3. Microfluidics in Sperm Counting

Swain et al. [69] explains a microfluidic chip fabricated on a glass wafer with channels to perform sperm counting; the system has two media reservoirs. Sperm movement was manipulated at different velocities by adjusting the height of the fluid columns from the two media reservoirs. Cells passed along a microfluidic channel that was flanked on either side with planar electrodes. The electrical impedance obtained was considered as sperm passed the electrodes. The sperm concentration was thus estimated. 

The paper discussed the physical principles of microfluidics along with the present designs and outcomes of microfluidic devices utilized for various steps that include gamete isolation and processing [70], fertilization and embryo culture in the embryo production in vitro (IVP) process. Gravity can serve to promote fluid flow via the hydrostatic pressure formed from varied heights of media columns. This approach was applied with micro-channel devices for use with embryos and sperm [71,72,73,74,75,76].

The recovery methods of sperm from men who undergo surgery due to non-obstructive azoospermia are outdated and may affect the quality of the sperm. A 3D-printed microfluidic chip to recover sperm from mixed cell suspensions, with a recovery rate higher than 96%, is explained in [77]. The various sperm selection strategies for the retrieval of high potential fertilizing spermatozoa and their impact on ART are explained in [78]. 

## 4. Microfluidics in Oocyte Sorting

A microfluidic device with three inlets and two outlets was fabricated for the separation of good quality bovine oocytes based on sedimentation rate differences in a sucrose buffer, which was dependent on oocyte quality, is explained in [79]. Figure 3A [79] shows the concept of oocyte separation in a microfluidic device. The sample solution was injected to the center inlet, which forms a confluent stream with the sucrose buffer injected from the other two outside inlets. After forming the confluent stream, the oocytes flowed through the separation channel and gradually settled. The end of the separation channel was divided into upper and lower outlets and connected to reservoirs. The good and bad quality oocytes were collected from the lower and upper outlet chambers, respectively.

In a di-electrophoresis (DEP) device, a healthy oocyte moves more rapidly than unhealthy oocytes. This hypothetical concept was experimented in a paper [80] on the DEP-based separation of healthy and unhealthy oocytes for IVF. Here, an electrode array chip of castellated shape was used to evaluate the di-electrophoretic velocities of oocytes. Based on the varied DEP response, a selected group of oocytes that moved towards the DEP electrodes showed a better developmental potential than the group of oocytes that stayed. The selected group of oocytes which moved to DEP electrodes showed a greater rate of blastocyst formation and a lower rate of polyspermy fertilization. [81]. 

Huang et al. [82] explains the trapping of mouse oocytes under a strong electric field that induced a p-DEP force between the gaps of the ITO-glass electrodes. Figure 3B [82] represents the oocyte trapping mechanism under an AC voltage at a frequency of the order of 1 MHz has been applied.

## 5. Microfluidics in Gametes Fertilization

The differentiation between microfluidic and traditional IVP along with the culture of single gametes/embryos is explained in [83]. Embryos cultured in a DMF chip with a PDMS ring have a significant probability of developing from a two-cell status into a blastocyst cell. The DMF manipulation of a droplet has hence solved gas exchange and is harmless to the embryos, which makes DMF chips biocompatible. Dynamic culturing powered with EWOD can manipulate a single droplet containing mouse embryos and culture to the eight-cell stage. The rates of IVF on the DMF platform and traditional Petri dish were 34.8% and 26.1%, respectively. About 25% of the embryos developed into the eight-cell stage; 33.3% of the embryos cultured in the Petri dish developed into the eight-cell stage [84].

### Microfluidic (Dielectrophoretic) System for Embryo Fertilization

DEP microfluidic devices that are fabricated with PDMS and indium tin oxide (ITO)-glass with electrodes, to capture and to screen the sperm in order to manipulate an oocyte, are explained in [85]. Oocytes subjected to an electric treatment had a developmental potential that was better than those without treatment. The paper discussed the enhancement of future fertilization in vitro rates for an oligozoospermia patient. Here, the sperm concentration near oocytes increased from 1.0 × 105 to 100 × 106 sperm/mL in the DEP microfluidic chip. Another related work stipulated the fertility rate in vitro in our DEP microfluidic chip to be 17.2 ± 7.5% at sperm number 3000, which was compatible with the standard rate of IVF, 14.2 ± 7.5%. 

The outcome measurements that imply ruminant embryos include rates of fertilization and cleavage (the proportion of inseminated oocytes undergoing at least one mitotic division), and blastocyst development (proportion of inseminated oocytes reaching blastocyst stage) are explained in [86]. Non-invasive morphological criteria assessed the blastocyst quality. During human reproduction, even the preimplantation embryo, including the morula and, later on, the early blastocyst, is subject to fluidic flow forces generated by peristalsis of the fallopian tube [87]. The use of microfluidic devices in the culture, maintenance and study of ovarian follicle development for experimental and therapeutic applications, along with microfluidic platforms for oocyte and sperm selection and maintenance, facilitation of fertilization with in vitro fertilization or intracytoplasmic sperm injection, and monitoring, selection and maintenance of resulting embryos, are explained in [88].

A positive DEP force is utilized in a PDMS microfluidic channel for the simultaneous manipulation and positioning of ICR mice oocyte and sperm to drive their position for natural fertilization. The rate of fertilization with the microfluidic chip in a DEP buffer solution was about 51.58% [81].

## 6. Microfluidics for Embryo Culturing

Embryo Culturing in a DMF EWOD platform is explained in [89]. IVF bypasses the female oviduct but it manually inseminates, fertilizes and cultivates embryos in a static micro drop containing appropriate chemical compounds. The dynamic culture powered by the EWOD system cultures a single embryo in vitro to mimic the in vivo environment, in a single droplet, in a microfluidic environment for developing the embryo to the blastocyst stage live births is given in Figure 4A [89]. The results show that the EWOD chip provides the culture of mouse embryos in a dynamic environment [89].

A microfluidic or micro-funnel culture system constructed of PDMS for mouse embryo culturing is explained in [90]. Here, the embryos were cultured on a flat and optically transparent floor (diameter 500 mm). The dynamic micro-funnel culture significantly improved embryo implantation and ongoing pregnancy rates over static culture.

A microfluidics platform enables embryo culture in precisely defined sub-microliter volumes (5–500 nL). Groups of two embryos per microfluidic well successfully developed to the blastocyst stage, at a rate of over 80%. Figure 4B [91] shows a microfluidic platform consisting of a vertically mounted chip that features an insertion funnel, a cylindrical culture chamber that can be closed off by two membrane valves, and an outlet port that performs embryo culture in sub-microliter volumes (5–500 nL) is explained in [91]. This design allows embryos to enter the culture chamber by gravity after being placed into the funnel by micropipette, thereby reducing any mechanical stresses that may compromise viability. Here, embryos were successfully developed to the blastocyst stage, at a rate of over 80%. 

Esteves et al. [92] explains the culturing of embryos in a confined microfluidic environment (fabricated in PDMS) as shown in Figure 4C [92] with nanolitre volume chambers that achieved pre-implantation developmental rates of up to 95% (4.5 days after fertilization) with normal birth rates (29–33%) under normal morphology resulting in full-term development after blastocyst transfer to the uterus. The refreshment of the dynamic culture medium during pre-implantation in a microfluidic system does not impair development to term.

Wang et al. [93] explains the adverse impact of women without functioning fallopian tubes in early 1978. During that time, women without functioning fallopian tubes were considered to be sterile by physicians. One patient’s fallopian tube is necessary to fertilize an oocyte by sperm in vivo. Earlier, many women with damaged tubes resorted to reparative surgery or tuboplasty in the hope of re-establishing a conduit for gametes to transit. Sadly, these surgeries often failed [93]. The story has changed progressively from there. Recent developments in bioengineering for the study of female reproduction, including bioengineering models of the ovary, fallopian tube, uterus, embryo implantation, placenta and reproductive disease, are explained in [94]. The microfluidic advancements in female reproduction are explained in [95]. 

## 7. Microfluidics in Embryo Quality Analysis

The evaluation of embryo quality, which is, in turn, helpful for assisted reproductive technology, is explained in [96]. Here, quantitative protein analysis has been implemented by the detection of human chorionic gonadotropin beta (hCG β) with a multicolor fluorescence detector (MFD). A spent embryo culture medium was used. Spent culture medium (SCM) has been proposed as an alternative source for embryonic DNA. Several studies have reported the detection of cell-free DNA in SCM and highlighted the diagnostic potential of non-invasive SCM-based PGT to assess the genetic status of preimplantation human embryos obtained with IVF [97]. Fluorescent magnetic beads (FMB) were used to capture hCG β, which β-galactosidase (β-Gal) was used to label. The method used a simple microfluidic chip and eliminated false positive signals generated with free β-Gal through simultaneous detection of the fluorescence. This method requires only 30-μL samples; the lower detection limit of hCG β was 0.1 pg/mL. hCG β secreted by top-quality blastocysts was greater than with embryos that failed to develop into a blastocyst and non-top-quality blastocysts. hCG β could, hence, be helpful as a biomarker to evaluate the quality of embryos. The identification and measurement, during IVF, of human embryo growth factors named human IL-1β and human TNF-α are explained in [98].

## 8. Microfluidic Diagnosis of Embryo Culture Medium

Dr. Wesley Kingston Whitten has made exceptional contributions towards the study of reproductive biology, and particularly of the preimplantation embryo. Dr. Wesley Kingston Whitten developed embryo culture medium, and so he is considered the ‘Father of Embryo Culture Medium’ [99]. A brief history of embryo culture medium is explained in [99]. The physical environment during in vitro embryo culture plays a major role in the quality of the resulting embryo. Understanding the pre-implantation embryo changes the philosophy of healthy human embryo culture, leading to a significant evolution in the culture media used in clinical IVF. [100,101,102,103,104]. Verification of these physical requirements through the development of novel culture platforms might lead to new future approaches, which in turn helps to improve ART. Static culture platforms culture the gametes and embryos on or in inert plastic vessels, ranging from test tubes to Petri dishes in various configurations. They lack an active means to agitate or to stimulate embryo or media movement. A dynamic culture platform treats the culture devices that are purposely engineered to stimulate controlled media flows. Criteria and considerations that must be fulfilled before any dynamic culture system receives widespread implementation were explained in [105]. 

The presence and extraction of cell-free DNA in an embryo culture medium is explained in [106,107]. In this work, a DMF EWOD magnetic bead (MB)-based DNA extraction was described; it demonstrated the extraction of cell-free DNA (cf-DNA) from an embryo culture medium of a mouse at a small concentration [106,107]. Here, KSOM served as the embryo culture medium. The magnetic property of DNA is explained in [108]. Alias A.B. et al. [106] explained the generation of a micro-droplet using DMF EWOD electrodes, as shown in Figure 5. The paper also discusses the extraction of small concentrations of cf-DNA from one micro-liter of mouse embryo culture medium with an EWOD platform. The magnetic behavior of DNA has been utilized for its extraction. Using magnetic beads, the cf-DNA from the culture medium was successfully extracted. 

## 9. Commercial Microfluidic Devices to Enhance Assistive Reproductive Technology

Over the past three to four decades, ART has expanded significantly around the globe. A commercially available ART IVF microfluidic device/kit is FERTILE Plus^®^ [109], which can be used in IVF (In vitro fertilization), IUI (Intra uterine insemination), and ICSI (Intracytoplasmic sperm injection). Fertile Plus provides sperm sorting, sperm washing, swimming and gradient centrifuging. Ref. [109] investigated whether Fertile Plus^®^ has a positive add-on effect on laboratory and clinical outcomes. In addition to this product, commercial male fertility test kits for personal home use, such as FertilMARQ31 or SpermCheck, are available to assess male fertility or male factor infertility in private. However, these test kits can only quantify sperm counts, not quantify sperm motility or the concentration of motile sperm [110]. A portable automated microfluidic device for rapid determination of sperm count is ‘Moxi Z’. This microfluidic device is an automated electronic cell counter which is utilized for determining sperm concentration [111]. 

Selective fractionation by density-gradient centrifugation is a kind of sperm separation. Percoll is a commercial medium for density-gradient centrifugation of cells, viruses and subcellular particles that is also widely used in IVF. The composition of Percoll is colloidal silica particles (15–30 nm in diameter) coated with nondialysable polyvinylpyrrolidone (PVP). Percoll density-gradient fractionation clearly separates spermatozoa from foreign material such as extender particles, cells and bacteria. The preparation of sperm for IVF was accomplished using the BoviPure^®^ gradient. It works at room temperature [112]. DxNow is a commercially available sperm selection method founded on the principles of microfluidics [33]. 

## 10. Supplementary Application of Microfluidics on ART

Along with the applications mentioned above, this paper reviews some of the microfluidic chip applications that mimic the mammalian reproductive system.

### 10.1. Microfluidics to Measure the Rate of Parthenogenesis in Mouse Oocyte

Parthenogenesis is a type of reproduction in which an egg can develop into an embryo without being fertilized by sperm. Spontaneous parthenogenetic and androgenetic events occur in humans, but they result in tumors [113]. The development of a parthenogenesis embryo of an infertile oocyte is similar to the development of an embryo in vitro, except for activation with various agents, such as electric or chemical agents, or stimulations of other types. The rate of parthenogenesis was measured on an ITO-glass electrode chip, with varied trapping duration and electric activation, under embryo formation from sperm and oocytes [82]. Here, the mouse oocytes were trapped with a positive di-electrophoretic force at the electrodes on an ITO-glass chip. This study demonstrated that parthenogenesis of the oocyte of an ICR mouse can be induced with an AC electric field and without sperm insemination.

### 10.2. Microfluidic Mammalian Oviduct

The oviduct is a convoluted tube that is composed of a longitudinal and circular muscular layer with a simple cuboidal to a columnar epithelium containing both ciliated and secretory cells [114,115,116]. A microfluidic IVF biochip system that imitates a mammalian oviduct to achieve enhanced rates of fertilization in vitro is explained in [81]; the microfluidic devices were fabricated with PDMS. Here, the simultaneous manipulation and positioning of oocyte and sperm of ICR mice through the microfluidic channel was achieved with a positive di-electrophoretic (DEP) force.

### 10.3. Microfluidics-Based Placenta-On-A-Chip

Placenta is a critical organ that supports the embryo growth by mediating the supply of nutrients and oxygen to a fetus as well as removing its waste products and carbon dioxide [117]. The significance of placenta is explained in [118,119,120] A microfluidic-based placenta-on-a-chip was developed with bioengineering techniques to simulate the placental interface between maternal and fetal blood in vitro and developed to analyze the transfer rate of glucose across the membrane [121].

## 11. Discussion

As ART, microfluidics has also rapidly evolved and diversified over the past three decades. The diversification from continuous microfluidics to droplet-based microfluidics enhances some unique characteristics of microfluidics as a high surface-to-volume ratio, ability to handle a small volume of fluids (microliter to Pico-liter), low reagent/waste requirements, fast response, the dominance of laminar flow, surface forces, molecular diffusion, and Brownian motion. These unique features, including portability and the ability to integrate multiple components on a single chip, make microfluidics an excellent tool for various fields of ART. Though ART procedures are mostly performed manually and require careful attention to detail and precision in handling and timing, this field of engineering has been successfully incorporated in various fields of ART. These ART fields can be categorized as (1) infertility diagnosis, (2) sperm selection, (3) sperm guidance, (4) oocyte analysis, (5) insemination, (6) embryo culture, (7) embryo selection, and (8) cryopreservation. Table 1 gives a brief idea about some of the various sperm sorting methods and their achieved efficiency.

PDMS is the widely accepted material for the fabrication of microfluidic devices due to its outstanding biocompatibility and simplicity of fabrication. The PDMS with glass bonds or PDMS with another PDMS layer bonding can achieve by a simple plasma treatment. Table 2 gives a brief overview of the various oocyte sorting methods.

However, self-regulation of the entire scope of ART procedures is yet to come. At present, the current potential for automation is found to be within the in-vitro fertilization laboratory [122]. A conceptual diagram of conventional and microfluidic IVF for ART is given in Figure 6.

## 12. Conclusions & Future of Microfluidic ART

The manuscript reviewed different aspects of microfluidic technology for ART applications based on the conceptual diagram shown in Figure 6. Over the last past decade, many significant advances have been made in the potential utility of microfluidics in the isolation, manipulation, analysis, and cryopreservation of gametes and embryos [26]. Though the basic science and technological demonstrations have been developed, the commercialization of microfluidic ART still has many hurdles to overcome. The microfluidic devices for sperm analysis and other IVF-on-a-chip technologies lack a commonly available interface. The high cost of laboratory accessory equipment such as syringe injectors, power sources, voltmeters makes another barrier to commercialization. Another reason for the slow takeoff of microfluidics from a broader commercial perspective is a poor understanding of the advantages afforded by microfluidics beyond workers in the field.

To overcome these hurdles, reproductive physiologists must work closely with engineers. Prototypes of such devices are currently being fabricated. Additional research is required to fully integrate the ART concept on a single microfluidic chip. Finally, the sterilization and packaging of the units must be redesigned and automated on a large scale to make them cheaply and easily available to many laboratories. 

## Figures and Tables

**Figure 1 molecules-26-04354-f001:**
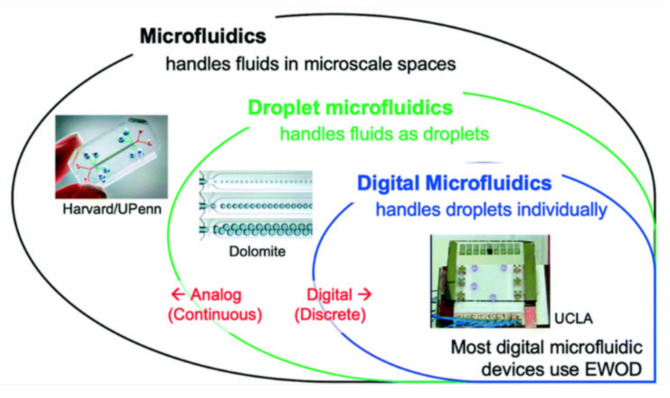
A road to analogue microfluidics to digital microfluidics. Image is adapted from ref. [25] with permission from the Royal Society of Chemistry.

**Figure 2 molecules-26-04354-f002:**
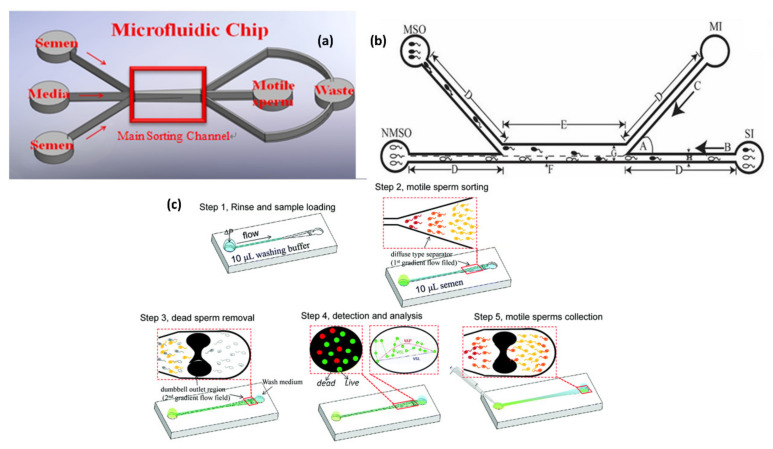
Different microfluidic devices for motile sperm sorting. (**a**) is adapted from ref. [61] with permission from Springer Nature, (**b**) is adapted from ref. [62] with permission from Springer Nature and (**c**) is adapted from ref. [63] with permission from Springer Nature and from the Royal Society of Chemistry.

**Figure 3 molecules-26-04354-f003:**
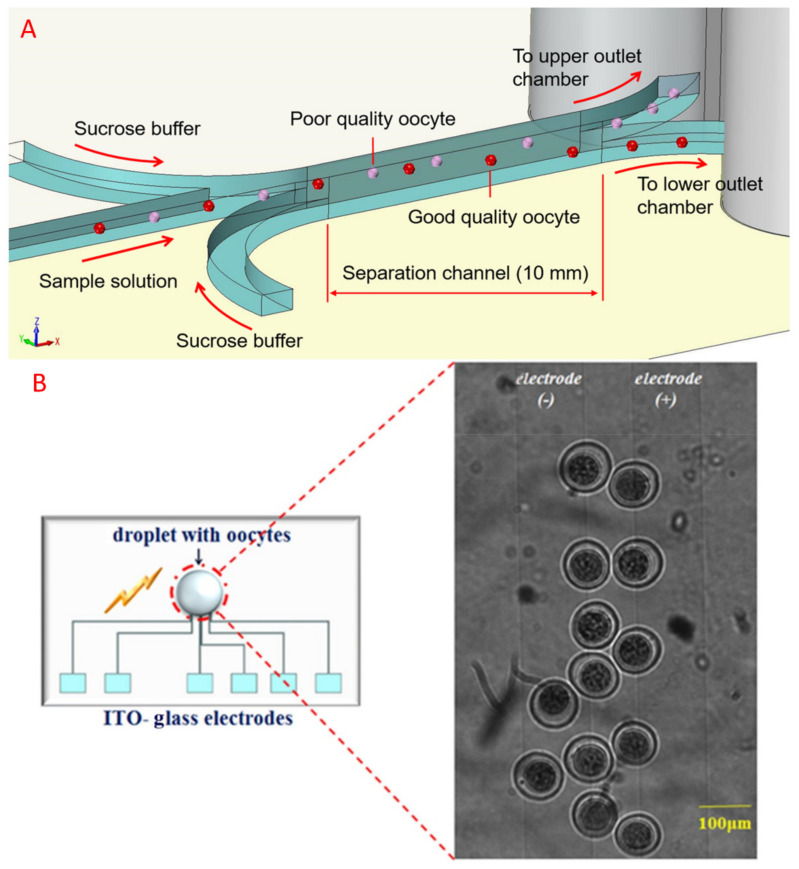
Various microfluidic device configurations to separate healthy oocytes. (**A**) is adapted from ref. [79] with permission from Springer Nature, and (**B**) is adapted from ref. [82] with permission from Wiley.

**Figure 4 molecules-26-04354-f004:**
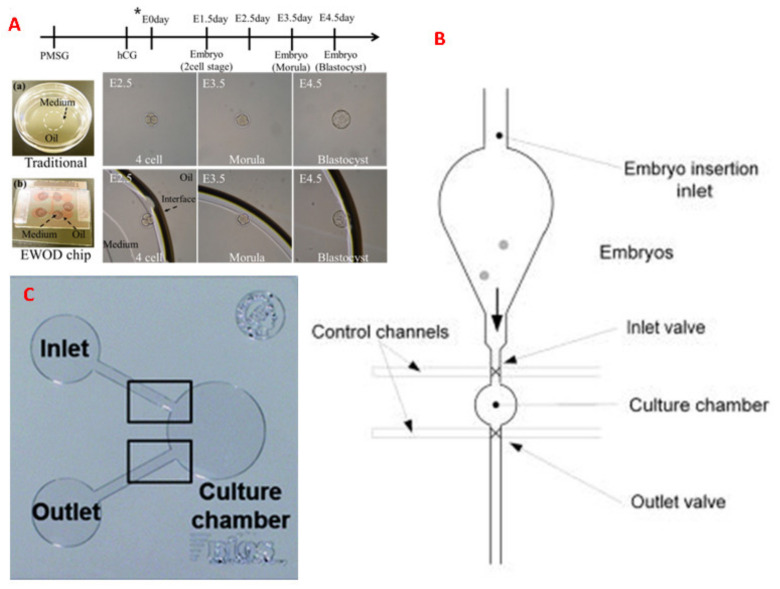
Different microfluidic device configurations for mouse embryo culture. (**A**) is adapted from ref. [89] with permission from PLoS ONE, (**B**) is adapted from ref. [91] with permission from John Wiley and Sons, and (**C**) is adapted from ref. [92] with permission from the Royal Society of Chemistry.

**Figure 5 molecules-26-04354-f005:**
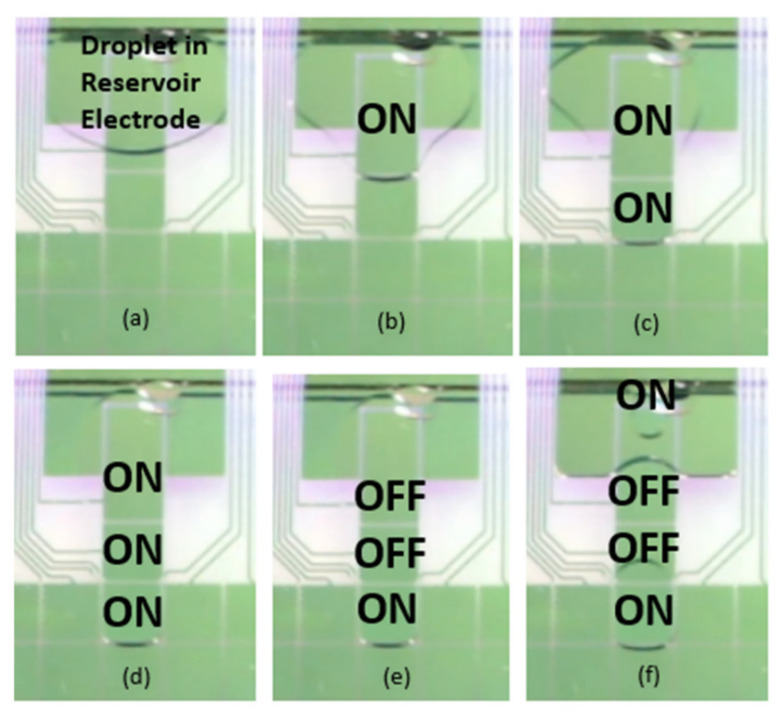
Real time discrete droplet generation in a DMF EWOD Chip. (**a**) represents the placement of droplet in reservoir electrode. In (**b**), generation electrode was turned ON. In (**c**–**e**), transportation electrodes were turned ON and OFF as per the quantity of droplet requirement. (**f**) shows the generation of droplet by turn ON the reservoir electrode. This figure is adapted from ref. [106] with permission from Springer Nature.

**Figure 6 molecules-26-04354-f006:**
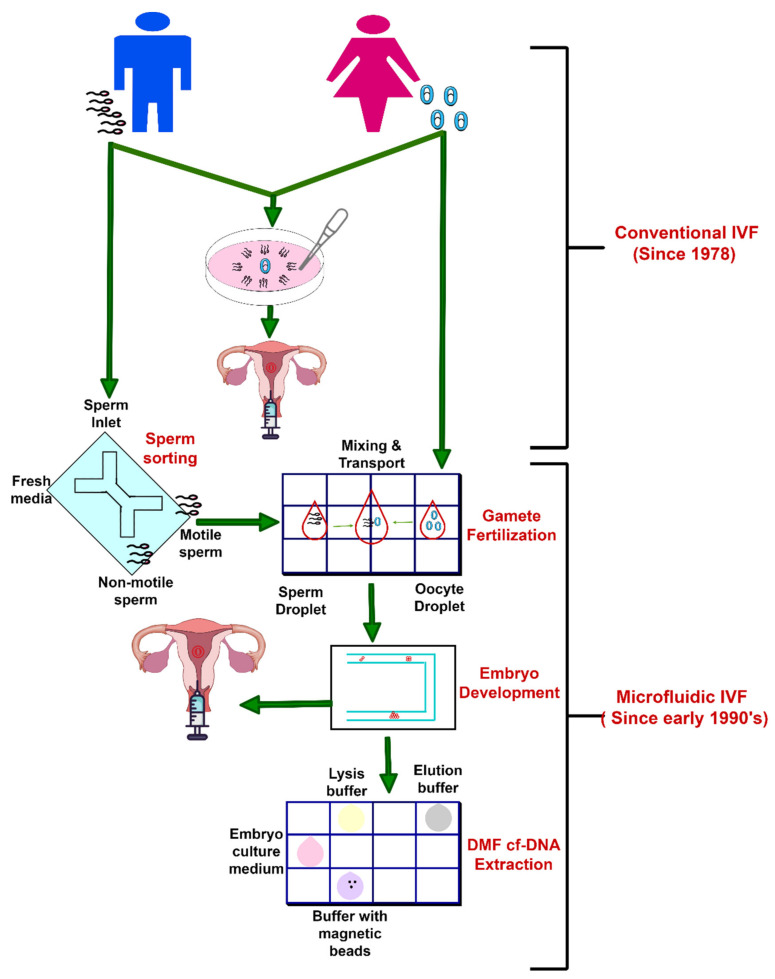
Conceptual diagram of conventional and microfluidic IVF for assistive reproductive technology.

**Table 1 molecules-26-04354-t001:** Various sperm sorting methods and their sorting efficiency.

Sl. No.	Material Used	Method Adapted	Efficiency	Ref. No.
1	PDMS	PDMS microfluidic device with three inlets and three outlets	90%	[61]
2	PDMS	Sperm-sorting test on microfluidic chip using optimal parameters	95.33%	[62]
3	PDMS	Flowing Upstream Sperm Sorter (FUSS)	90%	[63]

**Table 2 molecules-26-04354-t002:** Various oocyte sorting methods.

Sl. No.	Material Used	Method Adapted	Ref. No.
1	PDMS, PMMA	sedimentation rate differences in a sucrose buffer	[79]
2	Glass substrate, Gold Electrode	di-electrophoretic separation method	[80]
3	Glass substrate, ITO electrode	di-electrophoretic separation method	[82]

## Data Availability

Not applicable.

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
