# Peer review of "A Review on Microfluidics: An Aid to Assisted Reproductive Technology"

_molecules, 2021, doi:10.3390/molecules26144354_

Round 1

Reviewer 1 Report

In recent years, some comprehensive review articles about microfluidics and assisted reproductive technology have been published. Different microfluidic technologies enabled reproductive technologies were systematically summarized.

  • Thapa, S. and Heo, Y.S., 2019. Microfluidic technology for in vitro fertilization (IVF). JMST Advances, pp.1-11.
  • Weng, L., 2019. IVF-on-a-Chip: recent advances in microfluidics technology for in vitro fertilization. SLAS TECHNOLOGY: Translating Life Sciences Innovation24(4), pp.373-385.
  • Smith, G.D. and Takayama, S., 2017. Application of microfluidic technologies to human assisted reproduction. MHR: Basic science of reproductive medicine23(4), pp.257-268.
  • Sequeira, R.C., Criswell, T., Atala, A. and Yoo, J.J., 2020. Microfluidic systems for assisted reproductive technologies: advantages and potential applications. Tissue Engineering and Regenerative Medicine, pp.1-14.
  • Kashaninejad, N., Shiddiky, M.J.A. and Nguyen, N.T., 2018. Advances in microfluidics‐based assisted reproductive technology: From sperm sorter to reproductive system‐on‐a‐Advanced Biosystems2(3), p.1700197.

In this review article, the authors focus on the applications of droplet-based microfluidics on assisted reproductive technology. A systematic review is necessary of this field. However, clearly, this paper is a bit off-topic. Most works mentioned in this paper are not droplet-based microfluidics. This problem needs to be addressed to validify the rationality of this review. My additional comments are given below:

  1. The paper is not well organized.
    1. Sections 1-3 should be the introduction section and droplet microfluidics is not mentioned in the introduction.
    2. Sections 10 and 11 are confusing and there is no reason for introducing culture medium in a separated section.
    3. Section 12 simply lists recent advances which are not related to droplet microfluidics.
    4. It is not necessary to use subsections if there is only one subsection.
  2. In Figures 3 and 5, the caption is not clear, and the images are not well organised.
  3. This paper lacks sufficient latest references. Less than 15 research articles cited in this paper are published within the last 5 years.

Reviewer 2 Report

The paper covers several interesting aspects and applications of microfluidics in assisted reproductive technology. However, this paper only reports the information about these devices. Please see my comments below:

  1. Most of the devices discussed here are academic. It will be very useful if authors provide information about available commercial microfluidic devices for ART or comment on development of such microfluidic devices for ART. This can give an idea of where the field is headed.
  2. The paper does not have discussion section. The authors need to discuss their inferences based on the information they have provided and possible road map for future developments in this field. Also a note about the hurdles in the development should be mentioned. Also providing scope for future work can be helpful to the readers. Such information can give a clear picture about the field in this review.

Round 2

Reviewer 1 Report

The authors have reorganized the figures and the manuscript. The quality of the review paper has been improved. However, I could not see improvements on the following problems.

  1. Insufficient and out of date reference. The authors claimed that they have added recently published journals. However, I could not see much difference. And they cite less literature than the previous version. Could the authors respond clearly by highlighting recent research they added?
  2. The review topic is unclear. The authors added droplet microfluidics in the introduction but not much content could be found relating to ART. Also, in the discussion and conclusion part, droplet microfluidics is not mentioned. Considering this problem is not resolved after revision, is it possible for authors to revise the title of the paper?

Reviewer 2 Report

The authors have responded to the questions satisfactorily 

Author Response

There is no further questions.